# Hierarchical Nanocauliflower Chemical Assembly Composed of Copper Oxide and Single-Walled Carbon Nanotubes for Enhanced Photocatalytic Dye Degradation

**DOI:** 10.3390/nano11030696

**Published:** 2021-03-10

**Authors:** Kamal Prasad Sapkota, Md. Akherul Islam, Md. Abu Hanif, Jeasmin Akter, Insup Lee, Jae Ryang Hahn

**Affiliations:** 1Department of Chemistry, Research Institute of Physics and Chemistry, Jeonbuk National University, Jeonju 54896, Korea; mychemistry2037@gmail.com (K.P.S.); tina44445@gmail.com (J.A.); jrhstn@gmail.com (I.L.); 2Department of Chemistry, Amrit Campus, Tribhuvan University, Kathmandu 44618, Nepal; 3Department of Bioactive Material Sciences, Jeonbuk National University, Jeonju 54896, Korea; akherulraju@gmail.com (M.A.I.); hanif4572@gmail.com (M.A.H.); 4Textile Engineering, Chemistry and Science, North Carolina State University, 2401 Research Dr., Raleigh, NC 27695, USA

**Keywords:** CuO–SWCNT, nanocauliflower, heterojunction, recrystallization, photocatalysis, dye degradation, nanocomposites

## Abstract

We present the fabrication and proficient photocatalytic performance of a series of heterojunction nanocomposites with cauliflower-like architecture synthesized from copper(II) oxide (CuO) nanocrystals and carbon nanotubes with single walls (SWCNTs). These unique photocatalysts were constructed via simplistic recrystallization succeeded by calcination and were labeled as CuOSC-1, CuOSC-2, and CuOSC-3 (representing the components; CuO and SC for SWCNTs, and the calcination time in hours). The photocatalytic potency of the fabricated nanocomposites was investigated on the basis of their capability to decompose methylene blue (MB) dye under visible-light irradiation. Every as-synthesized nanocomposite was effective photocatalyst for the photodecomposition of an MB solution. Moreover, CuOSC-3 exhibited the best photocatalytic activity, with 96% degradation of the visible-light irradiated MB solution in 2 h. Pure CuO nanocrystals generated through the same route and pure SWCNTs were used as controls, where the photocatalytic actions of the nanocomposite samples were found to be remarkably better than that of either the pure CuO or the pure SWCNTs. The recycling proficiency of the photocatalysts was also explored; the results disclosed that the samples could be applied for five cycles without exhibiting a notable change in photocatalytic performance or morphology.

## 1. Introduction

Persistent, bio-accumulative, and toxic organic pollutants endanger human health via possible endocrine disruption, developmental defects, organ dysfunction, and chronic illness leading to death [1,2,3]. Their resistance to degradation via ambient physical, chemical, and biological actions favors their persistence in the environment [4]. Hence, researchers are extensively studying the degradation of such pollutants through photocatalytic oxidation reactions because of growing interest in sustainability. Photocatalytic oxidation reactions enable the direct storage of solar energy in chemicals, which is regarded as a sustainable approach [5]. Photocatalysis is regarded as one of the most promising approaches due to its capability to engender robust radicals in waterborne media by light irradiation; these radicals successively neutralize persistent pollutants thoroughly [6]. Low operational costs, facile reachability, outstanding performance, and potent action even in prevailing conditions of environment are additional advantages associated with photocatalysis [7].

Nanoscale semiconductor metal oxides and their nanocomposites have been widely investigated for their promising photocatalytic activity [8,9]. Nanosized copper(II) oxide (CuO) is appraised as a proficient photocatalyst on account of its great natural abundance, nontoxicity, excellent catalytic performance, robust chemical stability, and facile fabrication techniques [10]. In addition, CuO exhibits a contracted bandgap energy range from 1.2 to 2.6 eV; its particular bandgap energy is determined by the fabrication conditions. It comprises of *p*-type semiconductivity owing to the presence of oxygen vacancy defects [7]. In addition to its forefront application in photocatalysis, CuO has also been used in sundry technologies such as batteries, optoelectronics, gas detection, high-temperature superconductors, and heterogeneous catalysts [8,10,11]. However, CuO faces obstacles to its practical application as a photocatalyst. These problems include relatively weaker visible-light absorption, hindered accessibility to reaction sites, and the higher likelihood of electron–hole pairs recombination [7].

To overcome the shortcomings of CuO as a photocatalyst, nanoscale CuO can be chemically combined with other photoactive materials such as ZnO, ZnS, CdS, TiO_2_, SnO_2_, and carbon nanotubes (CNTs) [12]. Such chemical interactions give rise to substantial modifications of the electronic and geometrical structures of the composite product and improve its adsorption and dispersion characteristics [13]. The chemical combination of CuO nanoparticles and single-walled CNTs (SWCNTs) results in a composite with superior properties that make it suitable for use in multiple applications. Chemical bonds form between the outer walls of the SWCNTs and the CuO nanoparticles, leading to the formation of heterojunctions at their interfaces [14]. In addition, the SWCNTs are advantageous because of their plentiful desirable qualities, which include a larger aspect ratio, powerfully sharp tips, good thermal stability, high electron storage capability, superb electrical and thermal conductivities, and high mechanical strength [12,15].

Several researchers have reported improvements concerning the photocatalytic performance of CuO by forming heterojunctions with SWCNTs and multiwalled CNTs (MWCNTs). Chinnappan et al., for instance, fabricated three-dimensional (3D) hierarchically organized MWCNT/CuO nanostructures via a urea-supported chemical coprecipitation method and demonstrated it as an effective electrochemical catalyzer for the oxygen evolution reactions [16]. Khusnun et al. loaded CuO nanoparticles onto CNTs via an electrosynthesis method and reported an enhanced adsorption-oriented and visible-light-responsive CuO–CNT catalyst for the effective photodegradation of *p*-chloroaniline [17]. Mahmoodi et al. synthesized CuO/CNT nanocomposites through a wet chemical method and reported their enhanced photocatalytic degradation of dyes from colored textile wastewater [18]. Anderson et al. constructed catalysts for oxygen evolution via the loading of transition-metal oxides (including CuO) onto N-doped CNTs and reported enhanced catalytic performance [19].

In the present study, we designed and fabricated CuO–SWCNT (CuOSC) photocatalysts using a facile method to achieve desirable separation and immediate transfer of photogenerated electron–hole pairs, improved reception of visible-light, and outstanding activity. We designed and constructed different CuOSC nanocomposites while keeping the recrystallization parameters constant and altering the calcination time at a fixed temperature (550 °C). The nanocomposites were designated as CuOSC-1, CuOSC-2, and CuOSC-3 according to their constituents and their calcination time in hours. The photocatalytic competency of the devised nanocomposites was explored via their use for the ultimate neutralization of methylene blue (MB) dye under ambient radiation. The recyclability of the photocatalyst samples was assessed by their photocatalytic performance for five continual cycles. We observed no substantial change in the robust photocatalytic outcome or in the surface morphology of the catalysts after five cycles. The present work is the further exploration of our previous research [7]. The novelty of the current work includes an alteration in the synthesis method that allows the synthesis of more efficient photocatalyst with new architectures in significantly short time. Moreover, the novel hierarchical nanocauliflower chemical composites are effective as a photocatalyst in significantly lower dose.

## 2. Materials and Methods

### 2.1. Chemicals

Copper(II) acetate hydrate (Sigma-Aldrich, St. Louis, MO, USA, 98% purity), SWCNTs possessing an outward diameter of 1–2 nm, (US Research Nanomaterials, Inc., Houston, TX, USA, >90% purity), ethanol (Sigma-Aldrich, 99.5%), and MB (Alfa Aesar, Heysham, Lancashire, UK, high purity), potassium iodide (KI, Sigma-Aldrich, 99.99% purity), isopropyl alcohol (IPA, Sigma-Aldrich, 99.7% purity), 1,4-benzoquinone (BQ, Sigma-Aldrich, 99.99% purity) were employed with no additional purification; working MB solutions with the required concentrations were solubilized using distilled water.

### 2.2. Design and Construction of CuOSC Nanocomposites

Initially, a homogeneous solution was ensured by subjecting a mixture of copper(II) acetate hydrate (6.0 g) and ethanol (100 mL) contained in a hard-glass beaker to bath sonication for 50 min at room temperature (23 °C). SWCNTs (100 mg) were shifted into this prepared solution, and the resultant mixture was agitated magnetically for 1 h. The well-stirred mixture was kept untouched for recrystallization for 8 h after the magnetically stirring bar was drawn out. Crystals formed after the stirring bar was taken out; however, ~8 h was required for thorough and uniform recrystallization of the copper acetate around the SWCNTs. Recrystallized copper(II) acetate crystals–SWCNT agglomerates were set apart from ethanol (solvent) using vacuum filtration, and the residue obtained in filter the paper was dried at 70 °C in a programmable heating furnace for 2 h to completely evaporate the ethanol. The properly dried sample was transferred into a quartz crucible provided with a cover, and the loaded crucible was placed in a home-built stainless-steel chamber (SS316); the same chamber was subsequently sealed with a copper gasket and annealed in a furnace at 550 °C to fabricate the CuO and SWCNT heterocomposites. Different samples of CuOSC nanocomposites were synthesized at 550 °C, with only the calcination time varied. The nanocomposites obtained after 1, 2, and 3 h of calcination were named CuOSC-1, CuOSC-2, and CuOSC-3, respectively. The molar ratio computed as CuO-to-SWCNT in all of the synthesized CuOSC composite samples was 2.54:1 irrespective of the calcination time. The compositions were computed from the masses of SWCNTs taken for each set and the CuOSC composites obtained in three replicates of the experiments.

### 2.3. Characterization

The morphological features of the constructed nanocomposite samples were observed using field emission scanning electron microscopy (FE-SEM, SUB 8230, Hitachi, Chiyoda-ku, Tokyo, Japan) and high resolution transmission electron microscopy (HR TEM, JEM-2200FS, JEOL, Akishima, Tokyo, Japan), their crystallographic properties were evaluated by higher-resolution X-ray diffractometric analysis (XRD, Smart Lab, Rigaku, Akishima, Tokyo, Japan), their spectroscopic features were investigated using high-performance equipped X-ray photoelectron spectroscopy (XPS, K-Alpha, Thermo Scientific, Waltham, MA, USA) and UV–vis spectrophotometry, and their thermal stability was tested with a thermal probe device (SDT Q600 V20.9 Build 20, TA Instruments, New Castle, DE, USA). Brunauer–Emmett–Teller and Barrett–Joyner–Halenda (BET/BJH) measurements were carried out with an accelerated surface area and porosimetry system (ASAP 2420 V2.09, Micromeritics, Norcross, GA, USA).

### 2.4. Preparation of Pure CuO Nanocrystals

The same procedure described in Section 2.2 was used to prepare pure CuO nanocrystals out of copper(II) acetate hydrate and ethanol; the only difference from the procedure described in Section 2.2 was that we did not add SWCNTs. The thus-produced CuO nanocrystals were deployed to conduct control experiments to elucidate the photocatalytic propensities of the CuOSC based nanocomposites.

### 2.5. Photocatalytic Experiments

The photocatalytic experiments were carried out in different sets. For every set of experiments, an optimized doze (100 mg) of CuOSC photocatalyst was transferred to 100 mL MB dye solution blended in distilled water (final concentration of MB: 25 mg L^−1^). The homogeneous mixing of the components was ensured by sonication for 1 h; samples were kept untouched in a chamber for an additional 1 h to establish adsorption–desorption equilibrium among the dissolved MB entities and the photocatalytically hot locations. The photocatalytic investigations were performed on a cloudless sunny day under ambient conditions under sustained sunlight exposure between 12:00 pm and 2:00 pm; the outdoor temperature was between 20 °C and 23 °C, and the intensity of sunlight per unit area (average solar irradiance) was ~750 W m^−2^. A solar power meter (TM-206, Tenmars, Taipei, Taiwan) was used to record the intensity of solar radiation during the experiments. The photocatalytic performance was investigated by ensuring that only the visible and near-infrared components of sunlight were incident on the system; the UV component was selectively blocked from the incident radiation by employing a UV-cutoff filter. The CuOSC-catalyzed degradation of MB under natural sunlight was examined via an absorbance measurement using a UV–vis spectrophotometer.

### 2.6. Evaluation of Recycling Effectiveness

The recyclability of the CuOSC-3 photocatalyst was examined by gathering the used CuOSC-3 nanocomposite, washing it with distillate water, and drying it at 100 °C for ~20 min before reusing. The reuse activities were evaluated for five repeated cycles. The photocatalytic cycles were conducted between 12:00 p.m. and 2:00 p.m. for 5 sunny days at the unchanged experiment-spot to minimize variations in environmental conditions.

### 2.7. Determination of Point of Zero Charge and pH Effect Studies

We accessed the surface charge of the photocatalysts by determining their point of zero charge (pH_PZC_) value via pH drift technique. The pH_PZC_ value of the CuOSC nanocomposites was obtained by plotting the change in pH (ΔpH) against the initial pH (pH_i_). The initial pH of the test suspensions was adjusted employing 0.1 N HCl or 0.1 N NaOH. In each experiment set, 0.01 g of the photocatalyst was dispersed in 20 mL of pH adjusted 0.01 M KCl solution, stirred magnetically for 24 h and the final pH was recorded.

### 2.8. Charge Carriers Trapping Experiments

The scavenger agents, namely, isopropyl alcohol (IPA, 3 mL), potassium iodide (KI, 5 mg), and 1,4-benzoquinone (BQ, 5 mg) were employed in the experiment sets separately to trap the photon-induced hydroxyl (˙OH), holes (h^+^), and super oxide (˙O_2_^−^) radicals, respectively.

### 2.9. Liquid Chromatography–Mass Spectrometry (LC–MS) Analysis

The LC–MS technique was deployed to identify and confirm the degradation products of the MB solution. Each input comprised aliquots having 5 μL of refined sample injected into the LC system of LC 1200 Series (Agilent Technologies, Wilmington, NC, USA) equipped with a Phenomenex column maintained at 30 °C. The mass spectrometric characterization was carried out with an Agilent 6410B mass spectrometer (Agilent Technologies, Wilmington, NC, USA). The electrospray ionization (ESI) interface was used to generate ions in positive ionization mode. The ESI interface was steered at +3000 V with a source temperature at 380 °C, and the source offset, capillary voltage, and cone voltage were fixed at 30 V, 3 kV, and 30 kV, respectively. The other parameters were set at standard operating conditions. A single injection of sample was used for the experimental procedure, and the detection of ions was performed in the MS full-scan mode.

## 3. Results and Discussion

### 3.1. Morphological Properties

The morphological property of the as-constructed CuOSC nanocomposites was characterized by FE-SEM; illustrative micrographs are shown in Figure 1. The micrographs clearly show that the chemical combination of CuO and SWCNTs resulted in hierarchical growth of cauliflower-shaped composite particles with diverse sizes between 30 nm and 1 μm. All of the CuO crystals are nanodimensional; however, the sizes of the CuOSC nanocomposites appear to rely on the numbers of agglomerating crystals surrounding the SWCNTs. CuO nanocrystals encased the SWCNTs and resulted in the formation of nanocauliflowers. The space between consecutive nanocauliflowers reveals separation between the SWCNTs. Large-sized nanocauliflowers grew in areas that contained more SWCNTs, whereas voids were left in the areas without SWCNTs. SWCNTs are not distinct in the composites in Figure 1a–e due to hierarchical growth of the CuO encapsulating SWCNTs, however, the SWCNTs are apparently observable in Figure 1f (SWCNTs are revealed distinctly in the HR-TEM images, later in this section). The Figure 1f shows how CuO nanocrystals surrounded the SWCNTs to form chemically bonded nanocomposites. The size of the CuOSC nanocauliflowers appears to depend on the calcination time also. Bigger sized nanocomposites were observed in FE-SEM analysis of CuOSC-3 (Figure 1e).

The nanostructure of and chemical attachments between the constituents of the CuOSC composites were investigated by field emission energy-filtered TEM (FE-EF-TEM); representative micrographs are displayed in Figure 2. CuO nanocrystals are conspicuously attached to the walls of the SWCNTs; however, the nanocauliflower architecture observed by FE-SEM was dismantled by the bath sonication procedure used to prepare the TEM samples. CuO nanocrystals and SWCNTs combined chemically to generate an interconnected network of constituents. The chemical combination between the SWCNT wall and CuO nanoparticles gave rise to a permanent heterojunction between them; this heterojunction is a pivotal factor in the photocatalytic performance of the composite by promoting the split of photo-generated electron–hole pairs and preventing their recombination [5]. The CuO nanoparticles in the nanocomposites appear to possess monoclinic crystalline geometry. The well-distributed small nanocrystals exhibit sizes (as measured by FE-EF-TEM analysis) ranging from 2 to 15 nm. Since the TEM images of all the composite samples were almost similar, the representative micrographs of only CuOSC-3 nanocomposite are presented in Figure 2.

### 3.2. Structural Characterization

The crystallinity and the crystal structure of pure CuO, SWCNTs, and the CuOSC heterojunction-composites were examined by XRD analysis; the corresponding diffraction patterns are shown in Figure 3. The XRD profiles of the pure CuO (Figure 3a) include robust peaks at 2*θ* values of 43.34°, 50.53°, and 74.21°, which are attributable to the (111), (202), and (222) crystalline facets of monoclinic CuO [6]. All of the aforementioned peaks were also observed in the patterns of the CuOSC composite samples, along with a new peak due to pure SWCNTs. The XRD pattern of CuOSC-1 (Figure 3a) matches that of pure CuO except for the peak attributed to the (002) plane of the pure SWCNTs at 26.03°. However, two more peaks emerged at 36.42° and 61.46° in the pattern of CuOSC-2 (Figure 3c), corresponding to the (−111) and (−113) planes of monoclinic CuO, respectively [6]. Those two feeble peaks of CuOSC-2 became more prominent in the pattern of CuOSC-3 (Figure 3d), likely because of the longer calcination time of the CuOSC-3 nanocomposite.

The inset in Figure 3 shows the XRD pattern of untreated SWCNTs; the crests at 25.64° and 43.13° are attributed to the (002) and (100) crystal facets of graphitic carbon, respectively. The peak at 25.64° corresponds to the *d*-spacing of CNTs, confirming their crystalline form. The peak at 43.13° corresponds to disordered carbon present on the sample [20]. The peak corresponding to the (002) facet of crystalline carbon is observed in the patterns of all of the CuOSC nanocomposites, although it is shifted slightly toward higher 2*θ* values (approximately at 26°) compared with its position in the pattern of pure SWCNTs. A peak associated with the (100) plane of SWCNTs is not observable in the spectra of the nanocomposites because of its superimposition with the peak associated with the (111) plane of the monoclinic CuO. The presence of carbon peaks in the spectra of the composites suggests that the crystallinity of the carbon was unaffected by the nanocomposite formation process.

The Scherrer equation (Equation (1)) [17] was applied to obtain the mean crystallite size from full-width at half-maximum (FWHM) value determined from the XRD patterns of the nanocomposites:(1)$$D\;=\;\frac{k\;\lambda }{\beta \;\cos\theta }$$
where *D* is the mean diameter of the crystallites, *k* denotes the Scherrer constant (*k* ≈ 0.9), *λ* is the wavelength of the X-rays (*λ* = 0.154 nm for Cu Kα radiation), *β* is the FWHM in radians, and *θ* represents the diffraction angles in degrees. The peak at 43.34° (2*θ*) was chosen for reckoning the mean size, which was calculated to be 23.95 nm.

The charge states of the constituting elements in the samples were explored using high-performance equipped X-ray photoelectron spectroscopy (HP XPS); the corresponding spectra are depicted in Figure 4, and Appendix A.

The survey spectrum of the as-synthesized CuOSC nanocomposites is demonstrated by Figure 4a, where the presence of chemically combined Cu, O, and C is explicitly observed. The Cu-2*p* core-level profiles are displayed in Figure 4b. Deconvolution of the broad and asymmetric range reveals the coexistence of Cu^2+^ ions in neighboring different chemical environments. The peaks at 931.88 and 951.85 eV are indexed to Cu-2p_3/2_ and Cu-2p_1/2_ states, respectively and indicate the existence of Cu^2+^ in the CuOSC nanocomposite samples. The binding-energy gap of 20.03 eV, as obtained from a straightforward calculation using these values, coincides with the value reported in the literature, further confirming the Cu^2+^ state in the nanocomposites [21]. The satellite peaks at 943.70 and 962.17 eV confirm the purity of CuO in the composite samples [17].

The O-1*s* core-level details of the CuOSC-3 nanocomposite are illustrated by Figure 4c, where the broad asymmetric spectra have been deconvoluted into three discrete plots, revealing the existence of nonequivalent chemical atmosphere surrounding the O^2−^ ions. The profile with a peak at a binding energy of 530.26 eV affirms that the O^2−^ ions were connected chemically to Cu^2+^ ions in the monoclinic CuO, whereas the other element with a peak at 531.42 eV corresponds to vacant oxygen sites prevalent in the composite [17]. The XPS spectra reveal a substantial number of oxygen vacancies. The third and robust component at a relatively higher binding energy (533.13 eV) corresponds to chemisorbed O on the surface of the nanocomposite in the form of O_2_, H_2_O, or OH; such groups represent loosely attached oxygen species on the surface of the nanocrystalline CuO [10].

Deconvolution of the core-level C-1*s* profiles (Figure 4d) confirms the existence of dissimilar chemical environments around the carbon atoms in the CuOSC composite samples. The most intense peak at 283.98 eV indicates the dominance of *sp*^2^-hybridized graphitic carbon in the nanocomposites and that at 284.64 specifies defects in the form of sp^3^-hybridized carbon, whereas the peaks at 285.89 and 289.82 eV ascribe to C–O and O–C=O species, respectively [20]. The bonding states and chemical environments of the constituent elements, as previously discussed (bonding states of Cu, O and C discussed above), led us to conclude that heterojunctions formed between the CuO and SWCNTs via strong covalent bonds (explicitly, CuO–SC or Cu–OOC–SWCNT) or through linkage by means of van der Waals force.

### 3.3. Thermal Properties

The effect of thermal energy on the as-fabricated composite samples was evaluated using a thermal analyzer, in which the samples were heat-treated gradually from 27 °C to 800 °C sustaining a ramp speed at 10 °C min^−1^ under a nitrogen atmosphere. Thermogravimetric analysis (TGA) and differential scanning calorimetry (DSC) curves were constructed on the basis of the thermal analysis results; the corresponding spectra are displayed in Appendix A.

As expected for a stable metal oxide, pure CuO (Appendix A) exhibits a trivial mass loss when heated to 800 °C; the remnant mass of CuO after thermal action is 98.43%. A meager mass loss near 100 °C in the TGA trace of pure CuO is assigned to the evaporation of physically attached water molecules. The gradual decrease in mass between 545 °C and 785 °C is attributable to the thermal decomposition of residual acetate groups [7]. By contrast, the TGA trace of the SWCNTs (Appendix A) shows an enormous mass loss due to thermal decomposition; when the sample was heated to 555 °C, only 3% residual carbon remained; the remnant mass did not decrease further when the sample was heated to 800 °C. The slight weight loss observed in the temperature span of 100 to 390 °C is ascribed to the elimination of moisture content and to the deterioration of –OH groups attached to the SWCNTs. Afterwards, the fast loss between 400 and 550 °C is ascribed to the breakdown of SWCNTs [14].

The nanocomposite samples (Appendix A) demonstrated remarkably higher thermal stability than the pure SWCNTs. Almost no noticeable loss is observed in the TGA traces of the heterojunction composites heated to 500 °C; however, a slight mass loss occurred between 500 and 800 °C. Such a mass-loss trend, which is comparable to that of pure CuO but substantially greater than that of the SWCNTs, is ascribed to the complete removal of physisorbed water fragment, the degradation of bonded hydroxyl groups, and the decomposition of remnant acetates [14]. Among the nanocomposite samples, CuOSC-3 exhibited the greatest thermal stability, which is ascribed to its longer calcination time (3 h) compared with that of the other samples. The longer calcination time resulted in extensive removal of surface hydroxyl groups and unconsumed acetates. The results confirm that the thermal stability of the SWCNTs in the as-synthesized composites was dramatically enhanced by their chemical combination with CuO nanocrystals via strong chemical bonds. As validated by XPS analysis, the TGA results confirm that the attachment of CuO nanocrystals to the SWCNTs, as observed by FE-SEM and HR-TEM, occurs through chemical bonding between them. Such chemical bonds result in a permanent heterojunctions between their surfaces. The DSC profiles (Appendix A) affirm that the samples’ thermal events were all endothermic processes.

### 3.4. Optical Characteristics

The optical properties of the as-synthesized composites were examined by collecting their UV–vis diffuse-reflectance spectra and comparing them with those of pristine CuO; the corresponding spectra are revealed in Figure 5. The absorption range of pristine CuO is about 290 nm, which has increased gradually to reach the value of about 310 nm in the CuOSC-3 nanocomposite. Such a shift towards visible region compared to that of pure CuO improves the utilization of visible light. The absorbance spectra of separate samples were also utilized to draw Tauc plots to determine their bandgap energy. The Tauc plots (Appendix A) show that the bandgap value of the CuO (i.e., 1.75 eV) in the composite samples (1.62 eV in CuOSC-1, 1.54 eV in CuOSC-2, and 1.36 eV in CuOSC-3) was substantially lower than that of the pure CuO. This substantial depletion in the bandgap value of the CuO in the nanocomposites is accredited to the foundation of heterojunctions with the SWCNTs. Such a lower bandgap energy brings about enhanced generation of electron–hole pairs by the material under irradiation, resulting in greater photocatalytic efficiency toward decomposition of a substrate dye [21]. The CuOSC-3 exhibited the lowest bandgap energy among the prepared composites, indicating that it should exhibit the highest photocatalytic performance.

### 3.5. Surface Area and Volume Studies

The pore volumes and size spreads, and specific surface areas (SFA) of the samples were obtained using the BET/BJH methods; the main findings are summed up in Table 1. The results show that the SFA of the composite samples was greater than that of pure CuO and that the SFA of CuOSC-3 was the highest among the nanocomposite samples. We speculate that the chemical combination between CuO and SWCNTs resulted in an augmentation in the SFA of the as-synthesized samples. The higher the SFA of a photocatalyst, the greater the probability of adsorption of substrate molecules onto it, and thus the greater the likelihood of their photocatalytic degradation [17].

### 3.6. Photocatalytic Activity

The as-synthesized nanocomposites were tested for their photocatalytic proficiency by evaluating the catalyzed decomposition of MB under irradiation by natural sunlight, and their photocatalytic efficiency under identical conditions was compared with that of pristine CuO nanocrystals and SWCNTs. The obtained results are depicted in Figure 6. To investigate the possibility of self-degradation of MB alone under exposure to sunlight, we conducted a blank test by subjecting an MB solution under solar light exposure with no photocatalyst; the results are displayed in Appendix A. The results of the blank test show that natural sunlight alone (when no catalyst was used) caused negligible deterioration of the MB solution.

The photocatalytic performance of CuOSC-3, which reveals the highest photocatalytic performance among the investigated nanocomposite samples, is shown in Figure 6a; it exhibited 96% photo-decomposition of an MB solution through 2 h of sunlight exposure. The other samples (viz., CuOSC-2 and CuOSC-1) resulted in 93% and 90% photodecomposition at the same exposure time, respectively. By contrast, pure CuO and pure SWCNTs decomposed 56% and 11% of the MB solution, respectively, (Figure 6d and Appendix A, correspondingly) under the same experimental conditions and within the same time period. Appendix A demonstrates the photocatalytic action of pristine SWCNTs and the dose optimization of composite photocatalysts. The optimized doze (Appendix A) of photocatalyst, i.e., 100 mg photocatalyst per 100 mL substrate solution (1 g L^−1^) was employed in all experiment sets. These findings affirm that the establishment of heterojunctions between the crystalline CuO and the SWCNTs remarkably enhanced the photocatalytic ability of the CuO nanocrystals. Such an enhancement in photocatalytic performance is ascribed to the generation of an abundant number of charge carriers and their segregation due to synergistic interaction between the CuO and SWCNTs [16].

The progress of the photocatalytic decomposition reaction was monitored by plotting ln (*C*/*C*_0_) against the time of irradiation and the corresponding values are displayed in Figure 7a. The figure demonstrates that our as-synthesized photocatalysts are significantly better compared to the action of pristine CuO. We accessed the adsorption of dye on the surface of the photocatalyst by stirring the dye solution magnetically with the photocatalysts (1 h) and keeping undisturbed in a dark chamber (additional 1 h) for adsorption–desorption equilibrium. We found that MB was notably adsorbed on the surface of the photocatalyst during the adsorption–desorption equilibrium period. It is evident from 7a, where the concentration at 0 min is notably less than the initial concentration. There is the best adsorption of MB molecules by CuOSC-3 photocatalyst.

The kinetics of the photon-induced decomposition of MB via the photocatalytic action of as-synthesized catalysts was studied by calculating the decomposition rate constant with a pseudo-first order rate equation (Equation (2)):(2)$$\ln\;\left(\frac{C_{0}}{C}\right)\;=\;kt$$
where *k* is the first-order rate constant of the photodeterioration, *t* is the irradiation time in minutes, *C*_0_ is the initial concentration of the solution, and *C* is the concentration at the evaluated time [22,23]. The degradation rate constants calculated for CuOSC-3, CuOSC-2, CuOSC-1, and pure CuO samples were 0.0248, 0.0184, 0.0165, and 0.0082 min^−1^, respectively. Among the composite samples, the best photocatalyst, i.e., CuOSC-3, was examined for reusability and exhibited almost constant catalytic performance until the fifth reuse cycle; the photocatalyst decomposed 93% of the MB in the solution in its fifth recycle (Figure 7b). However, some changes in morphology due to repeated handling were observed (Appendix A). The recycling performances of CuOSC-1 and CuOSC-2 were not examined assuming that their abilities are comparable with that of CuOSC-3.

### 3.7. Effect of Point of Zero Charge and pH on Photocatalytic MB Degradation

The pH_PZC_ of the photocatalyst with the best activity was determined via pH drift technique and the obtained results are displayed in Appendix A; the pH_PZC_ value of the CuOSC-3 photocatalyst is 9.12. It implies that the surface of our as-synthesized photocatalysts is positively charged at pH value less than pH_PZC_ and positively charged at pH values more than pH_PZC_. Since MB is a cationic dye [24], there is an increase in the adsorption of MB on the surface of photocatalyst at pH higher than 9.12 and decrease in adsorption at pH less than that value. The effect of pH on the photodegradation of MB is presented in Appendix A. Lowest decomposition of MB was observed at pH 2; which is attributable to weak adsorption of cationic dye molecules on the photocatalyst surface due to stronger repulsion. Highest photodecomposition of MB was observed at pH 12 due to the high adsorption of cationic dye molecules on the surface of photocatalyst. In addition, excess hydroxyl ions (OH^−^) that are available at higher pH facilitate the generation of ˙OH radicals and enhance photocatalysis [24].

### 3.8. Mechanism of Photocatalytic Action

The mechanism of photocatalytic action of the nanocomposite samples can be explained based on the production of electron–hole pairs on the exposed surface of the photocatalyst being irradiated by sunlight comprising energy higher than or equivalent to the bandgap of the photocatalyst. In the nanocomposite photocatalyst, the energy of the radiation excites its filled valence-band (VB) electrons into the empty conduction band (CB), making photogenerated electrons (e^−^) available in the CB and generating positively charged holes (h^+^) in the VB [20]. In the hetero-combined CuO and SWCNTs, synergistic migration of electrons and holes is enhanced by the difference in their edge potentials; the VB edge potential of the CuO is lower than that of the SWCNTs, whereas it is exceedingly positive in comparison to the lowest unoccupied molecular orbital of the SWCNT [14]. Hence, the photogenerated (CB) electrons of the SWCNTs spontaneously drift towards the CuO CB and the photogenerated holes in the CuO VB correspondingly migrate to the SWCNTs through the heterojunction. Such an exchange of charge carriers between CuO and SWCNTs is responsible for their effective separation, resulting in a sufficiently large excess of photon-induced electrons and holes for initiating redox reactions that lead to the decomposition of dye solutions [6].

Oxygen vacancies (V_o_ in CuO) and the surface defects (SDs) between the CB and VB of SWCNTs (as observed via XPS analysis) in the hetero-composites further inhibit the recombination of photon-generated electrons and holes by ensuring their precise separation. The flow of photogenerated electrons from the VBs to the SDs and V_o_ regions and from these locations to the CB results in the substantial separation of charge carriers and promotes the generation of additional electrons and holes [17]. The photoinduced electrons attack dissolved O_2_ to generate superoxide radicals (˙O_2_^−^), which in turn produce powerful hydroxyl radicals (˙OH) that degrade MB molecules. The ˙OH radicals are primarily the prominent oxidizing agents that invade MB adsorbed onto the surface of the photocatalyst and generate transition intermediates [6]. The intermediates are repeatedly attacked by the ˙OH, ˙O_2_^−^, or h^+^ until they are converted into innocuous inorganic compounds such as H_2_O and CO_2_. The schematic diagram of the generation of reactive species is presented in Appendix A and the details of mechanism is provided thereafter in the Appendix A.

### 3.9. Charge Carriers Trapping Experiments

We performed charge-carriers trapping experiments to identify the main reactive species involved in the photo-degradation reactions and verify the mechanism of photo-deterioration of MB. The obtained results are displayed in Appendix A. The figure displays that ˙OH are the main reactive oxygen species, followed by ˙O_2_^−^ and h^+^, respectively, for the photo-deterioration of MB. The h^+^ seem to have least contribution for the photo-degradation process. The addition of IPA in the reaction mixture trapped the ˙OH radicals produced due to visible light irradiation upon the catalyst surface and suppressed the decomposition of MB. Similarly, KI trapped the h^+^ and BQ scavenged ˙O_2_^−^ to suppress the degradation reaction.

### 3.10. Detection of the Degradation Products

MB solutions before and after photocatalytic degradation were analyzed by LC–MS to characterize their photodecomposition products; the corresponding spectra are disclosed in Figure 8.

Figure 8a shows the LC–MS spectrum of an MB solution, which includes a robust peak at *m*/*z* 284.10; the detailed spectrum of the degradation products is shown in Figure 8b. Since the photocatalytic reactions with all as-synthesized samples were carried out under same experimental conditions and we expected the similar degraded products, we selected the degraded sample of MB by CuOSC-3 (sample with the best activity) for LC–MS analysis. The degradation intermediates or byproducts resulting from the photocatalytic action of our as-synthesized photocatalysts on MB were observed at *m*/*z* values of 145, 171, 203, 337, 365, 398, 421, and 463. Most of these *m*/*z* values of the degradation byproducts or intermediates are similar to those previously reported [25,26,27,28]. Because 96% degradation of the MB solution occurred during 2 h of catalyst exposure, some intact MB molecules were present in the solution, as evident in the LC–MS spectra. In addition, the intensity of the prominent MB peak in the LC–MS spectrum (Figure 8a) was substantially diminished in the spectrum of the photodegraded sample (Figure 8b). The structural formulae of the degradation products can be deduced on the basis of their *m*/*z* ratios. During the photocatalytic reaction, strongly oxidizing ˙OH radicals exfoliate the benzene rings of MB via demethylation or hydroxylation accompanied by oxidation [25,27]. A series of reactions occurs until the innocuous inorganic end products such as CO_2_, H_2_O, sulfates, and nitrates are produced from their organic parent molecules. Moreover, some nontoxic organic acids such as acetic acid and oxalic acid may also be produced. The LC–MS spectra confirm the photodegradation of MB and the formation of less-harmful organic or harmless inorganic products.

Based on the LC–MS analysis, the degradation mechanism of MB is proposed as revealed by Appendix A. Some of the additional intermediates (that are supposed to be vanished completely due to degradation) are also shown in the mechanism based on the literature for better understanding of the photodegradation mechanism.

## 4. Conclusions

An effective nanocomposite photocatalyst capable of degrading organic dyes under irradiation by natural sunlight was prepared employing a one-pot, straightforward recrystallization method succeeded by calcination. The nanocomposite demonstrated an impressive hierarchical nanocauliflower-like architecture composed of CuO nanocrystals chemically combined with SWCNTs. The results of XPS, HR-TEM, XRD, and TGA/DSC analyses confirmed the generation of heterojunctions between the outer walls of the SWCNTs and the CuO nanocrystals. BET/BJH and UV–vis absorbance measurements of the as-prepared photocatalysts showed desirable improvements in their surface area, texture, and visible-light absorption capability. The photocatalytic performance, as depicted via the UV–vis absorbance measurement of the substrate MB solutions versus exposure duration, revealed that the CuOSC-3 was the best photocatalyst among the as-synthesized samples. CuOSC-3 resulted in 96% photodegradation of MB in 2 h under sunlight exposure in outdoor environment conditions, whereas CuOSC-2 and CuOSC-1 resulted in 93% and 90% photodecomposition at the same exposure time under similar environmental conditions. The results affirm that our synthesized photocatalysts can function as efficient agents for the photocatalytic decomposition of persistent water pollutant organic dyes.

## Figures and Tables

**Figure 1 nanomaterials-11-00696-f001:**
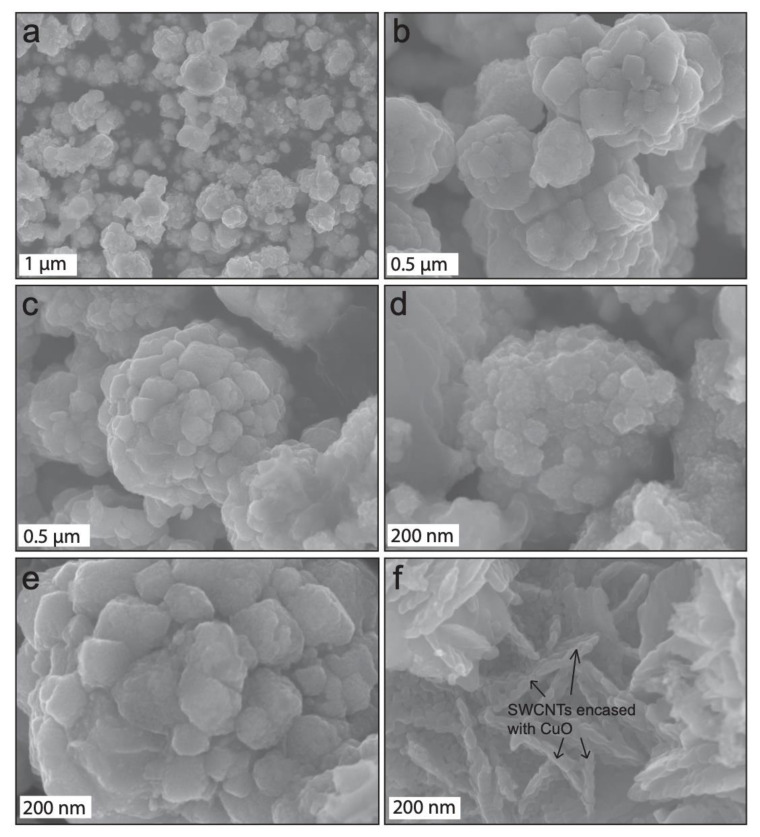
Representative field emission scanning electron microscopy (FE-SEM) micrographs of nanocomposites: CuOSC-1 (**a**) and (**b**), CuOSC-2 (**c**) and (**d**), and CuOSC-3 (**e**), exhibiting an interesting cauliflower-like architecture and (**f**) the chemical association of CuO and carbon nanotubes with single walls (SWCNTs) in the CuOSC-1 nanocomposites.

**Figure 2 nanomaterials-11-00696-f002:**
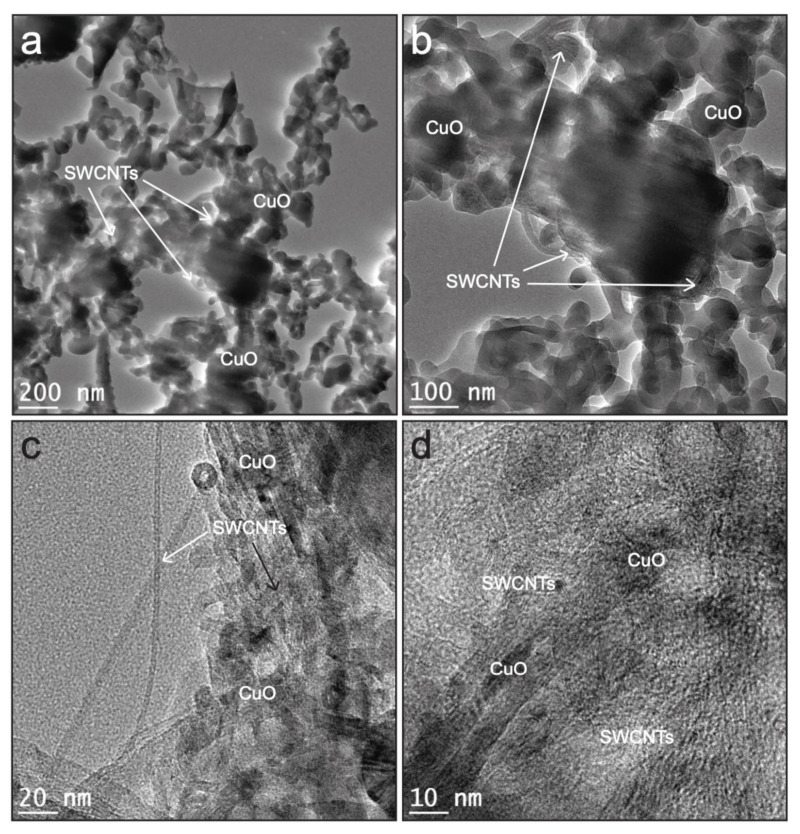
FE-EF-TEM images of CuOSC-3 nanocomposite displaying the attachment of CuO nanocrystals on the walls of SWCNTs via heterojunction formation: (**a**–**b**) interconnected networks of CuO and SWCNTs, (**c**) distribution of CuO and SWCNTs in the nanocomposite, and (**d**) lattice fringes of CuO and SWCNTs under high resolution.

**Figure 3 nanomaterials-11-00696-f003:**
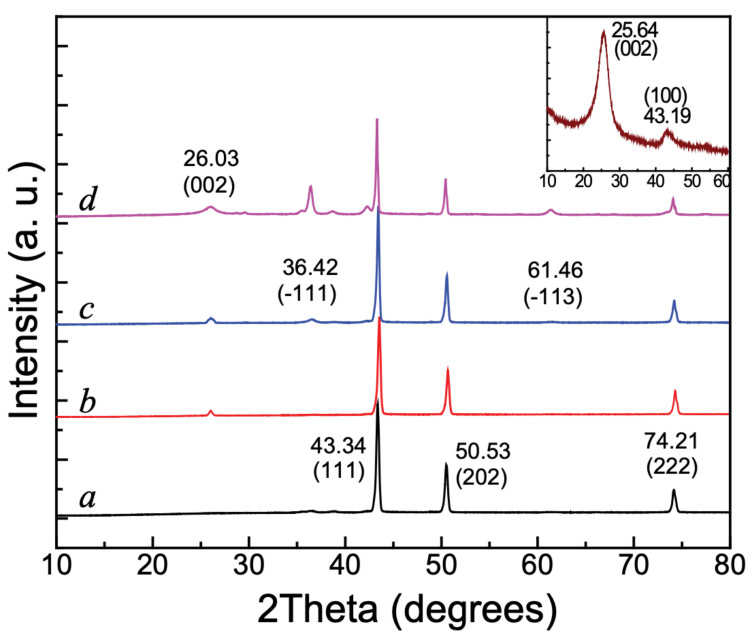
X-ray diffractometric analysis (XRD) plots of (**a**) pure CuO, (**b**) CuOSC-1, (**c**) CuOSC-2, and (**d**) CuOSC-3. The inset shows the XRD pattern of pure SWCNTs.

**Figure 4 nanomaterials-11-00696-f004:**
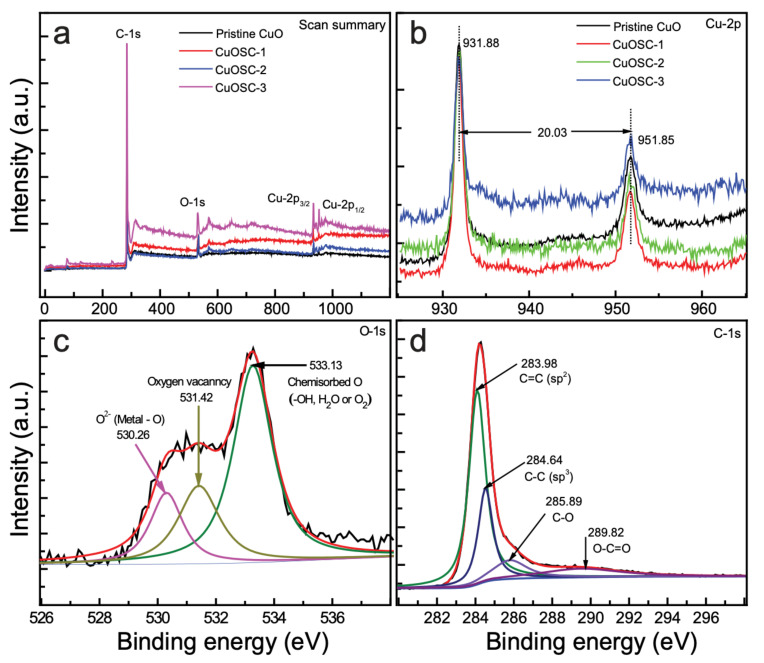
High-performance X-ray photoelectron spectroscopy (XPS) plots of the nanocomposites: (**a**) survey spectrum, (**b**) Cu-2*p* scan showing Cu-2*p*_3/2_ and Cu-2*p*_1/2_, (**c**) O-1*s* core-level of CuOSC-3, and (**d**) C-1*s* core-level spectra of CuOSC-3.

**Figure 5 nanomaterials-11-00696-f005:**
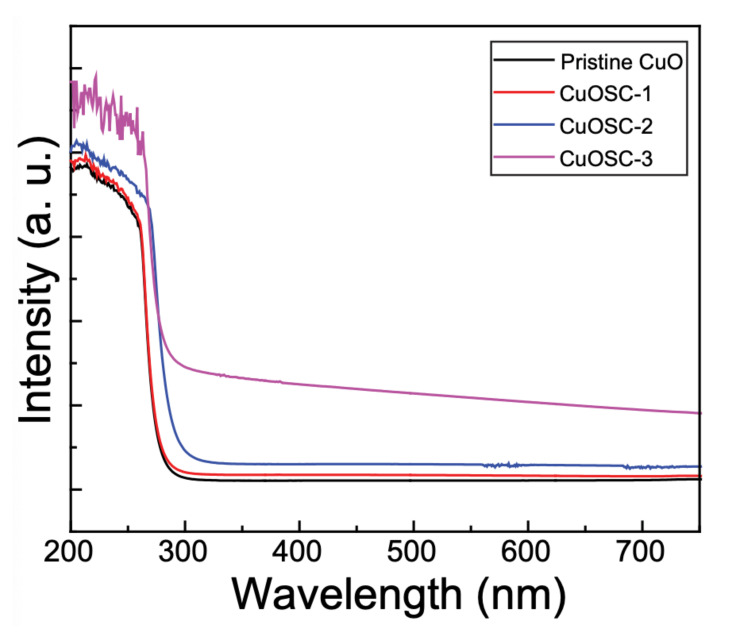
UV–vis diffuse-reflectance spectra displaying the improvement in optical absorbance range of pristine CuO owing to the formation of heterojunction with SWCNTs.

**Figure 6 nanomaterials-11-00696-f006:**
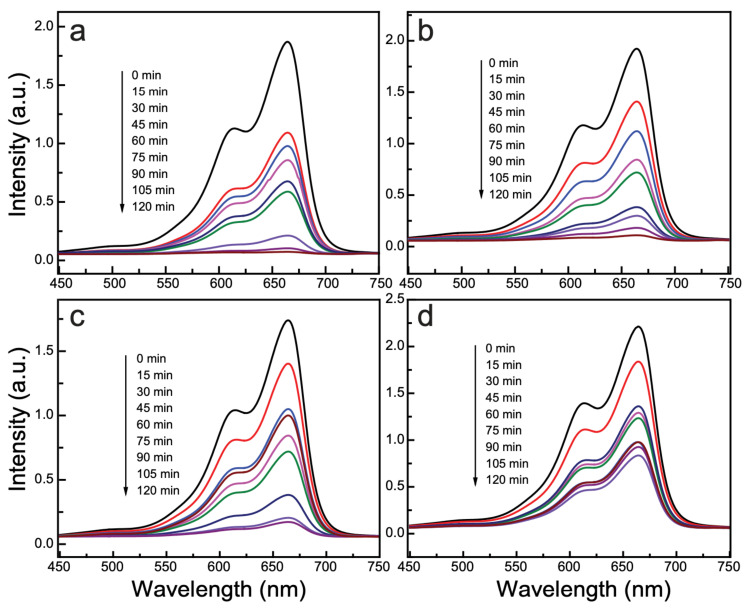
Photodegradation of the methylene blue (MB) solution varying with time of exposure, as examined by UV–vis absorbance spectroscopy: (**a**) CuOSC-3, (**b**) CuOSC-2, (**c**) CuOSC-1, and (**d**) pure CuO.

**Figure 7 nanomaterials-11-00696-f007:**
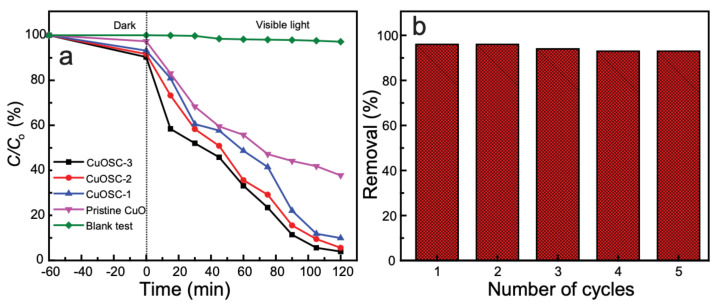
(**a**) Photocatalytic actions of different samples toward an MB solution and the results of a blank test; (**b**) recycling ability of the CuOSC-3 photocatalyst.

**Figure 8 nanomaterials-11-00696-f008:**
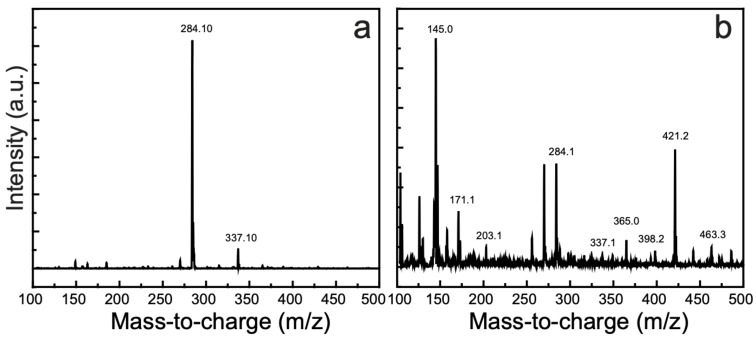
Mass spectra of an MB solution (**a**) before and (**b**) after the solution was subjected to photocatalytic action of CuOSC-3 for 2 h under solar-light irradiation. The full-scan spectra were recorded using a universal detector (+ESI scan (0.3 min), frag = 90.0 V, MB scan).

**Table 1 nanomaterials-11-00696-t001:** The pore volume, pore size distribution, and specific surface areas (SFA) of the samples.

Photocatalyst	*S*_BET_ (m^2^ g^−1^)	*V*_pore_ (cm^3^ g^−1^)	*D*_pore_ (nm)
CuO	33.96 ± 0.18	0.05 ± 0.001	3.36
CuOSC-1	35.07 ± 0.10	0.02 ± 0.0003	8.23
CuOSC-2	37.58 ± 0.18	0.04 ± 0.001	4.02
CuOSC-3	41.32 ± 0.19	0.11 ± 0.002	4.86
SWCNTs	258.84 ± 3.13	0.51 ± 0.002	8.45

*S*_BET_: BET specific surface area; *V*_pore_: total pore volume; *D*_pore_: average pore diameter.

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
