# Peer review of "Hierarchical Nanocauliflower Chemical Assembly Composed of Copper Oxide and Single-Walled Carbon Nanotubes for Enhanced Photocatalytic Dye Degradation"

_nanomaterials, 2021, doi:10.3390/nano11030696_

Round 1

Reviewer 1 Report

I accept all introduced changes. The manuscript may be published in a present form.

Author Response

Thank you very much for your acceptance.

Reviewer 2 Report

The resubmitted version of the manuscript that reports about the preparation of a series of heterojunction nanocomposites with cauliflower-like architecture synthesized from CuO nanocrystals and SWCNTs and their photocatalytic performance is considerably improved. The reviewers’ comments were appropriately addressed and the manuscript is more comprehensive now. Therefore, in my opinion it can be considered for publication. Though, I have a following suggestion: the excessive experimental details provided in the sections 3.7. Effect of Point of Zero Charge and pH on Photocatalytic MB degradation and 3.9. Charge Carriers Trapping Experiments should be presented in the 2. Materials and Methods section.

Author Response

Thanks for your kind comments. Please see the attached response.

Reviewer 3 Report

I believe that this revised version of the manuscript can be accepted for publication in this Journal.

Author Response

(The authors gave the same response as above.)

Reviewer 4 Report

Manuscript nanomaterials- 1128670 deals Hierarchical Nanocauliflower Chemical Assembly Composed of Copper Oxide and Single-Walled Carbon Nanotubes for Enhanced Photocatalytic Dye Degradation 4. The work reported a detailed characterization of the prepared materials, however, a considerable improvement can be taken into account in order to promote the quality of the work.

  1. Why you select a composition of 2.54:1 CuO-to-SWCNT? Did you optimize this composition? Did you prepare composites with other compositions?
  2. The catalyst concentration tested was f 1 g/L? This concentration is very high. Did you test other catalyst concentrations?
  3. The initial pollutant concentration is usually reported in g/L or mg/L instead of mg/mL, as reported here.
  4. What are the errors associated with SBET and Vpore values?
  5. The errors associated with analytical methods were also not considered. As no detection limits were reported.
  6. The catalytic activity of pure nanotubes was not presented. This result is mandatory to understand the presence of synergetic effect in the prepared composites.
  7. The catalytic activity was not evaluated in terms of organic matter removal. This is an important parameter to evaluate the performance of different tested samples. This parameter would also be considered in the cyclic experiments since the samples lose activity in terms of mineralization level along with reactions due to the adsorption of some by-products.

Author Response

(The authors gave the same response as above.)

Round 2

Reviewer 4 Report

The authors have taken into account the reviewers suggestions, therefore the paper can be published in the present form.

This manuscript is a resubmission of an earlier submission. The following is a list of the peer review reports and author responses from that submission.

Round 1

Reviewer 1 Report

The presented paper fulfills the Journal Scopus. The research objectives and results are clearly stated. The authors motivated by cited literature carried out experiments and confirmed known knowledge. However, in my opinion, the presented paper does not show the novelty in the area. Some studies need to be added. Additionally:

  1. The authors need to highlighted in the Introduction chapter, what is the novelty of this paper, or how does this research affect this field's knowledge?
  2. Zeta potential measurement – how was the dye chosen? Where there a specific analysis? Maybe this heterostructure is devoted to another dye groups?
    The adsorption of the dye at the surface of the photocatalyst. What about in this case? Were these changes observed? How they affect alumina nucleation? Was the pH effect analyzed?
  3. Figure 6, The authors analyzed only the direct allowed transition; however, it may not be so evident in the composite case. A further optical analysis is required
  4. The authors claimed that the interface between the CuO and the SWCNTs affects the photocatalytic adsorption, adjustment of some analysis (even theoretical will improve the paper).
  5. What about the inner particle diffusion model? What was the mechanism of the photocatalytic response? The presented sample exhibit a very complicated structure. Some insights will improve the presented data.
  6. According to the hydroxy radical species, a scavenger test will prove the author's theory. There are easy to do.

Recommendation Regarding This Manuscript: Reject

Reviewer 2 Report

The manuscript reports about the preparation of a series of heterojunction nanocomposites with cauliflower-like architecture synthesized from CuO nanocrystals and SWCNTs and their photocatalytic performance. The photocatalysts were obtained by recrystallization altering the calcination time at a fixed temperature (550°C). SEM analysis showed the hierarchical nanocauliflower-like architecture of the composites composed of CuO nanocrystals chemically combined with SWCNTs. The results of XPS, HR-TEM, XRD, and TGA/DSC analyses confirmed the generation of heterojunctions between the outer walls of the SWCNTs and the CuO nanocrystals. BET/BJH and UV–vis absorbance measurements evidenced improvements in the surface area, texture, and visible-light absorption capability of the nanocomposites. The photocatalytic efficiency of the nanocomposites was investigated by using methylene blue dye and VIS-light irradiation and it was found that all nanocomposites are effective photocatalysts. The recycling proficiency of the photocatalysts was also explored; the results disclosed that the samples could be applied for five cycles without exhibiting a notable change in photocatalytic performance or morphology.

The manuscript is well written, provides interesting results and can be considered for publication after addressing the following points:

  1. Section 3.1. Morphological properties. The SEM and TEM images analysis needs to be reconsidered because it is not clear what composites are illustrated in Figures 1 and 2. Moreover, the discussion should be done on all CuOSC-1, CuOSC-2, and CuOSC-3 composites in order to reveal the differences between their morphology and to further corroborate them with other structural particularities.
  2. Section 3.2 Structural Characterization. The XPS analysis needs to be reconsidered because it is not clear at all. Figure 4 illustrate the XPS slots of a composite, without being specified which one is it. The analysis must be performed for all investigated samples. Moreover, the asymmetric profile of the 931.81 eV peak illustrated in Figure 4b, must be explained.
  3. The authors examined for reusability only the CuOSC-3 sample. What was the reason to select only this sample? Please explain.
  4. Section 3.7 Detection of the Degradation Products. The authors are presenting and discussing only the mass spectra of a methylene blue solution before and after being subjected to photocatalytic action of the CuOSC-3 compound for 2 h under solar-light irradiation. What about the spectra of other compounds? They should be also presented or at least discussed throughout this section.
  5. There are some typing mistakes that need to be corrected.

Reviewer 3 Report

The Authors report  the synthesis and  photocatalytic activity of a series of nanocomposites with cauliflower-like architecture synthesized from copper(II) oxide  (CuO) nanocrystals and single-walled carbon nanotubes.

The paper is well written and the reported issues have relevance in photocatalysis and in environmental field. However, I believe that before the manuscript can be considered for publication in this or other Journals, the novelty of the work with respect to what reported by the same Authors in a very similar work must be defined. In fact, in a previous study (see ref. [7] of the manuscript), the Authors have already reported the synthesis of a series of copper(II)-oxide-single-walled carbon nanotube composites and have investigated their photocatalytic performance by evaluating the degradation of methylene blue.

Reviewer 4 Report

The manuscript entitled Hierarchical Nanocauliflower Chemical Assembly Composed of Copper Oxide and Single-Walled Carbon Nanotubes for Enhanced Photocatalytic Dye Degradation reported the preparation of Cu/SWCNT composites for dye degradation by photocatalysis. I consider that this work needs major revision since mandatory parameters were not considered during the study. Additionally, the novelty carried out with this work would be emphasized, as additional comments would be taken into account before publication, once the work is very descriptive and almost no discussion is reported.

  1. In general, the work only reported the results, and no discussion was performed. Therefore, I suggest that the paper needs an extension revision in order to clarify the obtained results.
  2. The work presents many figures and some of them are unnecessary. For example, Figures 5 and 6 can be removed. The data of these figures can be only reported during the manuscript.
  3. In figure 8 the xx and yy axes would initiate at 0, 0.
  4. The results of LC-MS are presented in Figure 9, but no information is discussed. The authors wrote that only the dye is degraded in several by-products, but this is only informative.
  5. The reusability tests only reported the dye removal, and no correlation was performed with these results.